# Benzoic Acid Derivatives of *Ifloga spicata* (Forssk.) Sch.Bip. as Potential Anti-Leishmanial against *Leishmania tropica*

**Syed Majid Shah** [1,2], **Farhat Ullah** [1,*], **Muhammad Ayaz** [1,*], **Abdul Sadiq** [1], **Sajid Hussain** [1,2], **Azhar-ul-Haq Ali Shah** [3], **Syed Adnan Ali Shah** [4,5], **Nazif Ullah** [6], **Farman Ullah** [2], **Ikram Ullah** [7] and **Akhtar Nadhman** [8]

[1] Department of Pharmacy, University of Malakand, Chakdara, Khyber Pakhtunkhwa 18800, Pakistan; majidpharma08@yahoo.com (S.M.S.); sadiquom@yahoo.com (A.S.); hussain77pk2003@yahoo.com (S.H.)
[2] Department of Pharmacy, Kohat University of Science & Technology, Kohat 26000, Pakistan; pharmankhan@yahoo.com
[3] Department of Chemistry, Kohat University of Science & Technology, Kohat 26000, Pakistan; izhar_hej@yahoo.com
[4] Faculty of Pharmacy, Universiti Teknologi MARA Puncak Alam Campus, Bandar Puncak Alam, Selangor 42300, Malaysia; benzene301@yahoo.com
[5] Atta-ur-Rahman Institute for Natural Products Discovery (AuRIns), Universiti Teknologi MARA Puncak Alam Campus, Bandar Puncak Alam, Selangor 42300, Malaysia
[6] Department of Biotechnology Abdul Wali Khan University, Mardan 23200, Pakistan; ullahnazif@awkum.edu.pk
[7] Suleiman Bin Abdullah Aba-Alkhail center for Interdisciplinary Research in Basic Sciences, International Islamic University, Islamabad 46000, Pakistan; ikram.ullah@iiu.edu.pk
[8] Institute of Integrative Biosciences IIB, CECOS University, Peshawar 25000, Pakistan; shamsnazman@gmail.com
* Correspondence: farhataziz80@hotmail.com (F.U.); ayazuop@gmail.com (M.A.); Tel.: +92-333-936-1513 (F.U.); +92-346-800-4990 (M.A.)

**Abstract:** This study aimed to appraise the anti-leishmanial potentials of benzoic acid derivatives, including methyl 3,4-dihydroxybenzoate (compound **1**) and octadecyl benzoate (compound **2**), isolated from the ethnomedicinally important plant *Ifloga spicata* (*I. spicata*). Chemical structures were elucidated via FT-IR, mass spectrometry, and multinuclear ($^1$H and $^{13}$C) NMR spectroscopy. Anti-leishmanial potentials of the compounds were assessed using *Leishmania tropica* promastigotes. Moreover, acridine orange fluorescent staining was performed to visualize the apoptosis-associated changes in promastigotes under a fluorescent microscope. A SYTOX assay was used to check rupturing of *Leishmania* promastigote cell membranes using 0.1% Triton X-100 as positive control. A DNA interaction assay was carried out to assess DNA attachment potential. AutoDock software was used to check the binding affinity of compounds with surface enzyme leishmanolysin gp63 (1LML). Both compounds exhibited considerable anti-leishmanial potential, with $LD_{50}$ values of $10.40 \pm 0.09$ and $14.11 \pm 0.11$ μg/mL for compound **1** and compound **2**, respectively. Both compounds showed higher binding affinity with the leishmanolysin (gp63) receptor/protease of *Leishmania*, as assessed using computational analysis. The binding scores of compounds **1** and **2** with target gp63 were −5.3 and −5.6, respectively. The attachment of compounds with this receptor resulted in their entry into the cell where they bound with *Leishmania* DNA, causing apoptosis. The results confirmed that the investigated compounds have anti-leishmanial potential and are potential substitutes as natural anti-leishmanial agents against *L. tropica*.

**Keywords:** *Ifloga spicata*; benzoic acid derivatives; *Leishmania tropica*; docking; DNA interaction; apoptosis

## 1. Introduction

Leishmaniasis is a major health threat affecting more than 350 million people across the globe and resulting in 70,000 deaths per year [1–4]. It is caused by an intracellular protozoa parasite belonging to the genus *Leishmania* [5]. Cutaneous leishmaniasis, mainly initiated by *Leishmania tropica*, is most common in the Northern Areas of Pakistan, and is spreading throughout the country due to the migration of several million refugees [6,7]. This condition is usually self-healing, within 3–18 months, but may sometimes be chronic, and the wounds leave disfiguring spots which lead to public separation and mental pressure. In local communities, herbal therapies are preferred, owing to their safety, efficacy, fewer side effects [8,9], and the unavailability of standard therapies. The rational search for new anti-leishmanial agents can replace the need for current chemotherapies, including those based on antimonial and amphotericin B, which have many undesirable side effects [10–12].

Agathisflavone, camptothecin, quercetin, and sinefungin are phytochemicals which have been isolated from different medicinal plants and have exhibited promising anti-leishmanial potential [13]. Moreover, the investigated compound, methyl 3,4-dihydroxybenzoate, has been isolated from plants *Piper glabratum*, *P. acutifolium*, and *Asphodelus microcarpus* [14–16], and was found to be active against different strains of *Leishmania*. However, we are reporting, for the first time, the efficacy of methyl 3,4-dihydroxybenzoate and octadecyl benzoate against *L. tropica*, the most prevalent species of *Leishmania* in Pakistan. The work is further extended to investigate the preliminary leishmanicidal mechanism of these natural compounds.

Molecular docking studies are an important tool for the discovery of anti-leishmanial compounds [17]. Leishmanolysin (gp63) is a membrane-bound glycoprotein present on the promastigote surface of various species of *Leishmania* protozoans, and has a key role during *Leishmania*-induced infection; thus, it can be an attractive target for the discovery of novel anti-leishmanial drugs. [18]. SYTOX assays are used to characterize the membrane distraction potential of substances, as SYTOX dye can pass through the ruptured plasma membrane but cannot cross undamaged membranes, thus distinguishing between dead and living cells [19]. Furthermore, DNA studies are important to check the contact of the agents with *Leishmania* DNA because damage to DNA halts the replication/transcription process, leading to apoptotic death of *Leishmania* [20]. Apoptosis is a physiological procedure of cell suicide and in *Leishmania* species it initiated in response to various anti-leishmanial drugs and nutritional deficiencies [21–24].

*Ifloga spicata*, which belongs to the family Asteraceae, is an annual herb used as whole plant for the treatment of skin diseases like dermatosis and allergies, and in decoction form for cardiac diseases [25–27]. Approximately 20 different species of the Asteraceae family have been found to possess anti-leishmanial potentialities [28]. The purpose of this study was to isolate and purify the compounds and evaluate their anti-leishmanial potentials against *L. tropica*, along with the possible mechanism.

## 2. Materials and Methods

### 2.1. General Experimental

NMR spectrometer, Bruker AVANCE 500 MHz, was used to record the $^1$H and $^{13}$C spectra. IR and, mass (EI and HERI-MS) spectra were recorded on a JASCO A-302 and variant mass spectrometer, respectively. Extraction was carried out with methanol (Daejung CAS No.67-56-1) while chromatographic isolation was performed with the help of organic solvents *n*-hexane (Daejung CAS No.110-54-3), chloroform (Daejung CAS No.67-66-3), and ethyl acetate (Daejung CAS No.141-78-6). Thin-layer chromatography was performed on a silica gel 60 F254 card (Merck, EMD Millipore

Corporation, Billerica, MA, USA), while for column chromatography, silica gel (230–400 mesh) was used. Compound spots were observed with ultraviolet light and stained by spraying with a solution of ceric sulphate. The *L. tropica* cultures were maintained in Medium 199 (M199) (Gibco, Invitrogen, Carlsbad, CA, USA) supplemented with 10% heat-inactivated fetal bovine serum (PAA Laboratories, GmbH, Austria), 100 U/mL penicillin (Sigma, Milwaukee, WI, USA), and 100 mg/mL streptomycin (Bio Basic Inc., Markham, ON, Canada). An ELISA reader (ELx800 BioTek) was for used for MTT assays, while a UV–vis spectrophotometer (Shimadzu, Kyoto, Japan, UV-1800) was used for the DNA interaction study. A fluorescent microscope (Leitz Laborlux) was used to observe apoptosis and membrane permeability.

## 2.2. Plant Material, Extraction, and Fractionation

*I. spicata* mature plant was collected at the flowering stage in the start of April 2016 from Karak District, Khyber Pakhtunkhwa, Pakistan, and its identification was confirmed by taxonomist Dr. Waheed Murad, Department of Plant Sciences, Kohat University of Science and Technology (KUST), Kohat. A voucher specimen was deposited in the department herbarium with a voucher number (KUH 1002). The whole plant (leaves, flowers, stem, roots) was collected, shade-dried for 15 days, and coarsely crushed using a cutter mill. Maceration was carried out for the powdered plant (18 kg) in 80% methanol followed by filtration. The filtrate was concentrated at 40 °C using a rotary evaporator to obtain methanolic extract (600 g with yield 3.8%). The crude extract (500 g) was dispersed in distilled water (500 mL) and sequentially extracted with *n*-hexane, chloroform, ethyl acetate to obtain polarity-based fractions [29–32]. All fractions were subjected to preliminary anti-leishmanial screening, and the ethyl acetate fraction was found to be the most effective.

## 2.3. Bioguided Isolation and Characterization

Column chromatography was performed for the most bioactive ethyl acetate-soluble fraction using a mixture of *n*-hexane/ethyl acetate as the mobile phase, and flash silica gel as the stationary phase. The five eluted sub fractions with mobile phase *n*-hexane/ethyl acetate were obtained. The two eluted sub fractions, fraction 1 and fraction 2, were subject to repeated column chromatography using a pencil column. Compound **1** (15 mg) was obtained from fraction 1 using mobile phase *n*-hexane/ethyl acetate in the ratio 90:10, and compound **2** (12 mg) from fraction 2 using *n*-hexane/ethyl acetate in the ratio 94:6. The structure of the compounds was confirmed through various spectroscopic methods, like $^1$H and $^{13}$C techniques, including mononuclear (COSY) and heteronuclear correlation experiments (HSQC and HMBC), and a literature survey [33–35].

## 2.4. Anti-Leishmanial Activity against Leishmania Promastigotes

Compounds were dissolved in dimethyl sulfoxide (DMSO) using a concentration of 1 mg/mL to for stock solutions, which were successively further diluted. About 180 µL of M199, 100 µL of *L. tropica* log phase culture, and 20 µL of each compound were added to the wells of a microtiter plate and incubated at 24 °C for 72 h. The viability of the *L. tropica* promastigotes was observed by the formation of purple formazan crystals in the living cells via mitochondrial dehydrogenase enzyme. The percentage of promastigote survival was measured quantitatively at 540 nm using a microplate ELISA reader, with Glucantime as a positive control [36].

## 2.5. Molecular Docking of Leishmanolysin (gp63)

AutoDock software was used to analyze the binding affinity of test compounds with surface enzyme leishmanolysin gp63 (PDB code: 1LML) in the form of E-values. 2D structures of test compounds were converted into 3D structure format by using Biovia Discovery Studio Visualizer 2016 client. This software was also used for post-docking analysis and schematic representation of the hydrophobic interactions, number of hydrogen bonds, and amino acid residues involved in the best-docked pose of the compound–receptor complex [37].

## 2.6. Mechanistic Anti-Leishmanial Studies

### 2.6.1. SYTOX Assay

This assay was used to check *Leishmania* promastigotes cell membrane rupturing by using 0.1% Triton X-100 as a standard. Promastigote culture was added to the 96-well plate, incubated at 24 °C, and centrifuged at 13,000 rpm for 10 min. Cells were suspended, washed with Hank's buffer salt solution (HBSS) three times, and were resuspended in HBSS along with 1 mM SYTOX Green stain. The plate was kept in the dark for 10–15 min, and a fluorescent microscope was used to check the membrane permeability [38].

### 2.6.2. DNA Extraction

DNA was extracted from *Leishmania* by adding 100 µL of lysis buffer and incubating in a water bath at 60 °C for 2 h. Organic solvents were added to the above solution and centrifuged at 10,000 rpm for 10 min. Precipitation of DNA occurred via addition of 1 mL of chilled isopropanol to the aqueous layer and, again, centrifuged using the same conditions. DNA was washed with 70% ethanol and suspended in Tris-EDTA buffer.

### 2.6.3. DNA Interaction Assay

The assay was carried out by dissolving both compounds and DNA in double de-ionized water (pH = 7.0). *Leishmania* DNA concentration was measured via absorption spectroscopy, using the molar absorption coefficient of 6600 $M^{-1}$ $cm^{-1}$. The UV absorption titrations were performed with and without DNA. Compound–DNA solutions were kept for 10 min at room temperature, and then a reading was taken [20].

### 2.6.4. Apoptosis Analysis Studies

Acridine orange fluorescent staining assay was performed to visualize the apoptosis-associated changes in *L. tropica* promastigotes under a fluorescent microscope (Leitz Laborlux). Promastigotes maintained and cultured in RPMI-1640 were incubated in 96-well plates ($1 \times 10^4$ cells/well) at 25 °C ± 1 in the presence and absence of both compounds. The cells were harvested and checked for apoptosis-associated changes after 48 h. After the respective periods of incubation were completed, a 10 µL suspension of the treated leishmanial cells were transferred to glass slides, to which 1 µL (100 µg/mL) of acridine orange was added and the entire suspension was covered with a cover slip. The changes in the morphology and nuclear staining of the cells were observed under a fluorescent microscope coupled with a camera [39].

## 2.7. Statistical Analysis

Experimental results were expressed as mean ± standard deviation and repeated three times. Two-way analysis of variance (ANOVA) and *t*-tests were performed to determine significant means. GraphPad Prism 5 software was used to calculate $LD_{50}$ values.

## 3. Results and Discussion

Medicinal plants are playing a significant role in drug discovery and development against numerous disorders [40–44]. The main advantage of medicinal plants as a source of new drugs is the molecular diversity of secondary metabolites in these herbs [45,46]. In the current study, compound **1** (Figure 1) was obtained as amorphous powder. The molecular formula was determined to be $C_8H_8O_4$, based on HREIMS and $^{13}C$ NMR spectra. The $^1H$ NMR spectrum displayed signals for three aromatic protons, suggesting a tri-substituted benzene ring. The three substituents were easily assigned as two hydroxyl groups and a carbonyl group, keeping in view the molecular formula, $C_8H_8O_4$; resonances in the IR spectrum, 3338, 3186, and 1662; and signals in $^{13}C$ NMR at δ 166.9, 148.5, and 143.0 ppm.

The appearance of two aromatic protons at δ 7.61, 7.57 ppm showed their presence at ortho positions due to the electron deshielding effect of the carbonyl group supported by HMBC. In addition, all the three protons were chemically and magnetically non-equivalent, reflecting the two hydroxyl groups at the 3 and 4 position of the benzene ring. Proton NMR (500 MHz, CDCl$_3$) signals appeared at 7.61 (1H, s, H-2), 7.57 (1H, d, $J$ = 8 Hz, H-6), 6.90 (1H, d, $J$ = 8 Hz, H-5), 3.89 (3H, s, OCH$_3$), while carbon $^{13}$NMR (125 MHz, CDCl$_3$) signals appeared at 166.9 (C=O), 148.5 (C-4), 143.0 (C-3), 123.7 (C-6), 122.7 (C-1), 116.6 (C-2), 114.8 (C-5), and 51.9 (OCH$_3$) (Figures 1–3). The data were consistent with previously published data [16,47] (Supplementary Materials Figures S1–S3).

**3,4-Dihydroxy-benzoic acid methyl ester**

**Benzoic acid octadecyl ester**

**Figure 1.** Chemical structures of the isolated compounds.

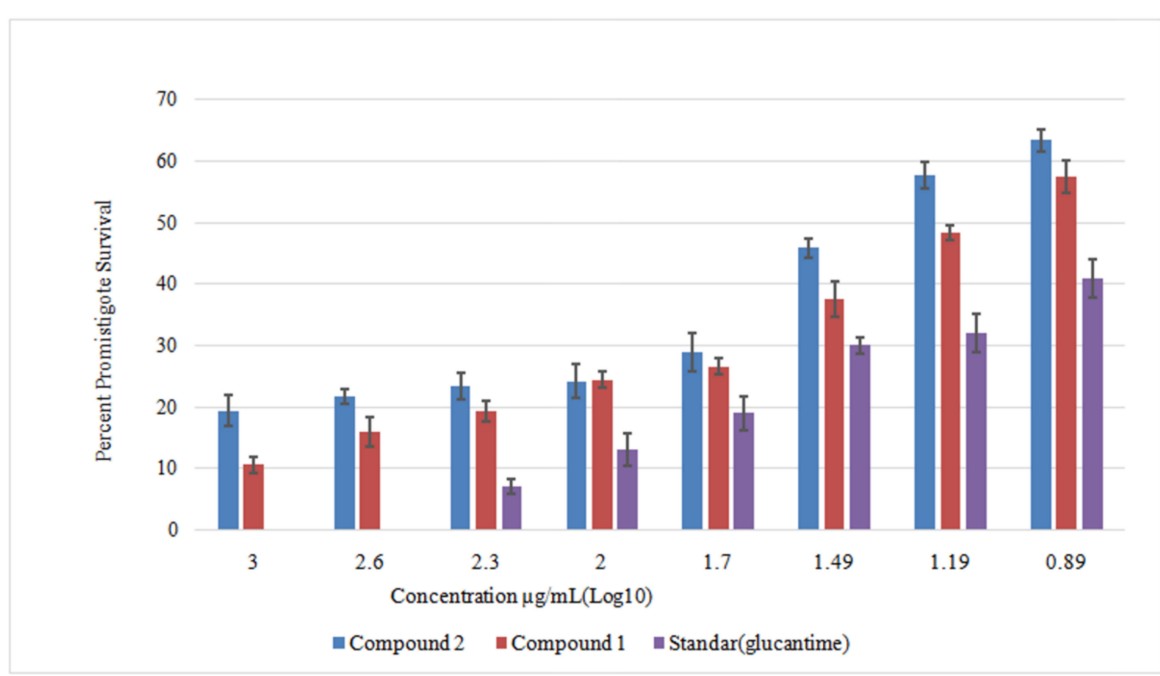

**Figure 2.** Anti-leishmanial potential of compound **1** and compound **2** in a dose-dependent assay, depicting promastigote survival after 72 h of treatment. Bars show significant difference ($p$ < 0.05) in survival rates of the parasites of respective groups. Standard drug; Glucantime LD$_{50}$ = 5.33 ± 0.07 μg/mL.

The structure of compound **2**, octadecyl benzoate (Figure 1), was elucidated with the help of IR, EI, NMR spectroscopy (including 2D NMR techniques), and by comparison with the literature [48]. It was obtained as colorless amorphous powder. The molecular formula was determined to be C$_{25}$H$_{42}$O$_2$, based on HREIMS and $^{13}$C NMR spectra. $^1$H NMR spectrum displayed a triplet at δ 0.88 and a broad

singlet at δ 1.26, typical of a straight chain hydrocarbon. The $^{13}$C NMR spectrum (BB and DEPT) showed 25 signals comprising 1 methyl, 17 methylene, 5 methine, and 2 quaternary carbons. The signal in the downfield region at δ 1672 was assigned to the carbonyl carbons. The oxygenated methylene resonated at δ 65.1. The absence of methine and quaternary carbon in the upfield region suggested that the alkyl chain was linear. This could be further confirmed by the presence of terminal ethyl group (triplet at δ 0.88 J = 7.8 Hz and multiplet at δ 1.26). The benzene ring was found to be monosubstituted due to the presence of three signals each in $^1$H NMR (8.04 (m, 2H, 2′, 6′), 7.4 (m, 1H, 4′), 7.2 (m, 2H, 3′, 5′), and $^{13}$C NMR 130.6 (C-1′), 129.5 (C2′, 6′), 128.3 (C-3′, 5′) spectra. Proton NMR (500 MHz, CDCl$_3$ signals appeared at 0.88 (3H, t, J = 7.8 Hz), 1.26 (m, (CH$_2$)9), 1.53 (m, CH$_2$), 1.76 (m, CH$_2$), and 4.3 (m, CH$_2$) while $^{13}$C NMR (125 MHz, CDCl$_3$) appeared at 167.0 (C=O), 132.7 (C-4′), 130.6 (C-1′), 129.5 (C2′, 6′), 128.3 (C-3′, 5′), 65.1 (C-1), 32.0 (C-2), 29.7 (C-3-11), 29.6 (C-16), 29.5 (C-13), 29.3 (C-12), 28.7 (C-14), 26.0 (C-15), 22.7 (C-17), and 14.1 (CH$_3$) (Figures 4 and 5 and (Supplementary Materials Figures S4 and S5)).

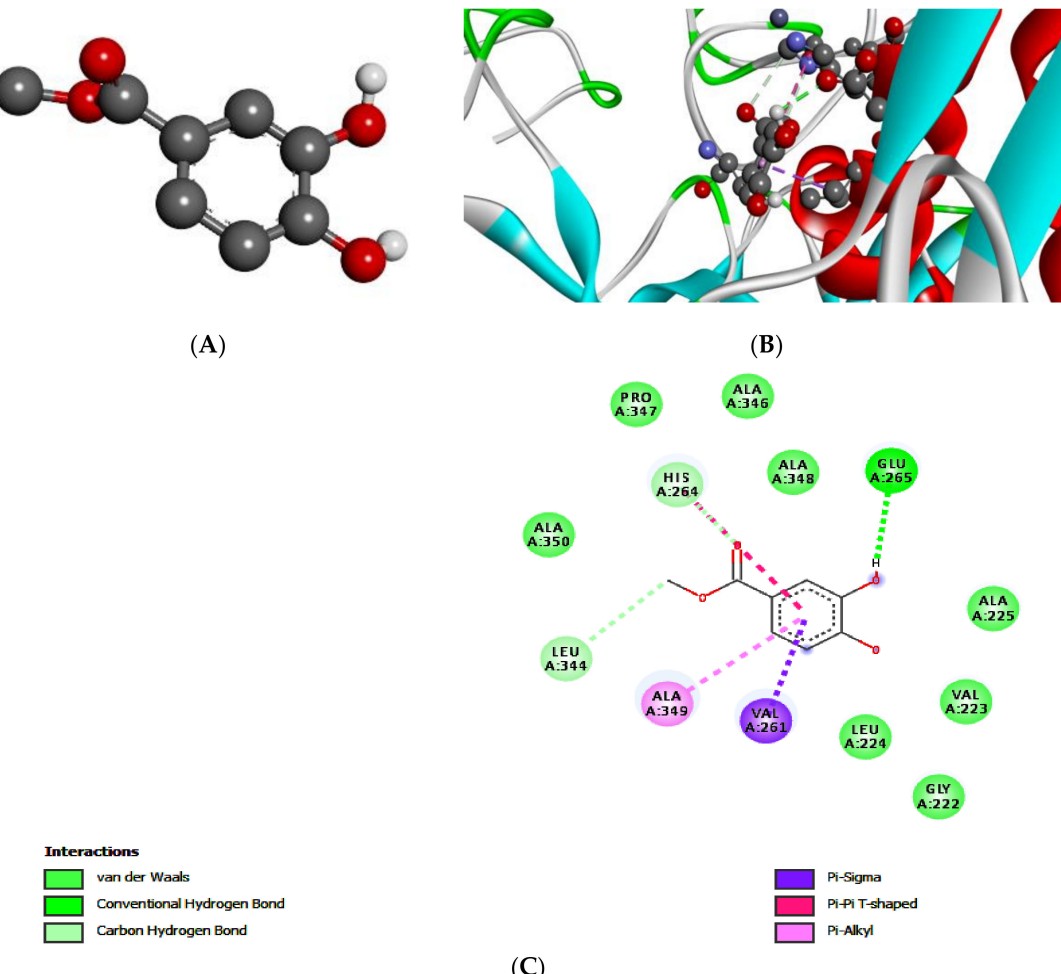

**Figure 3.** Interactions of legend (compound **1**) with *Leishmania tropica* promastigote surface gp63 enzyme, drawn through Discovery Studio Visualizer client 2016. (**A**) PDB structure and best pose of methyl 3,4-dihyroxybenzoate; (**B**) General interaction of methyl 3,4-dihyroxybenzoate with its targets; (**C**) Specific interaction of compound **1** with its targets.

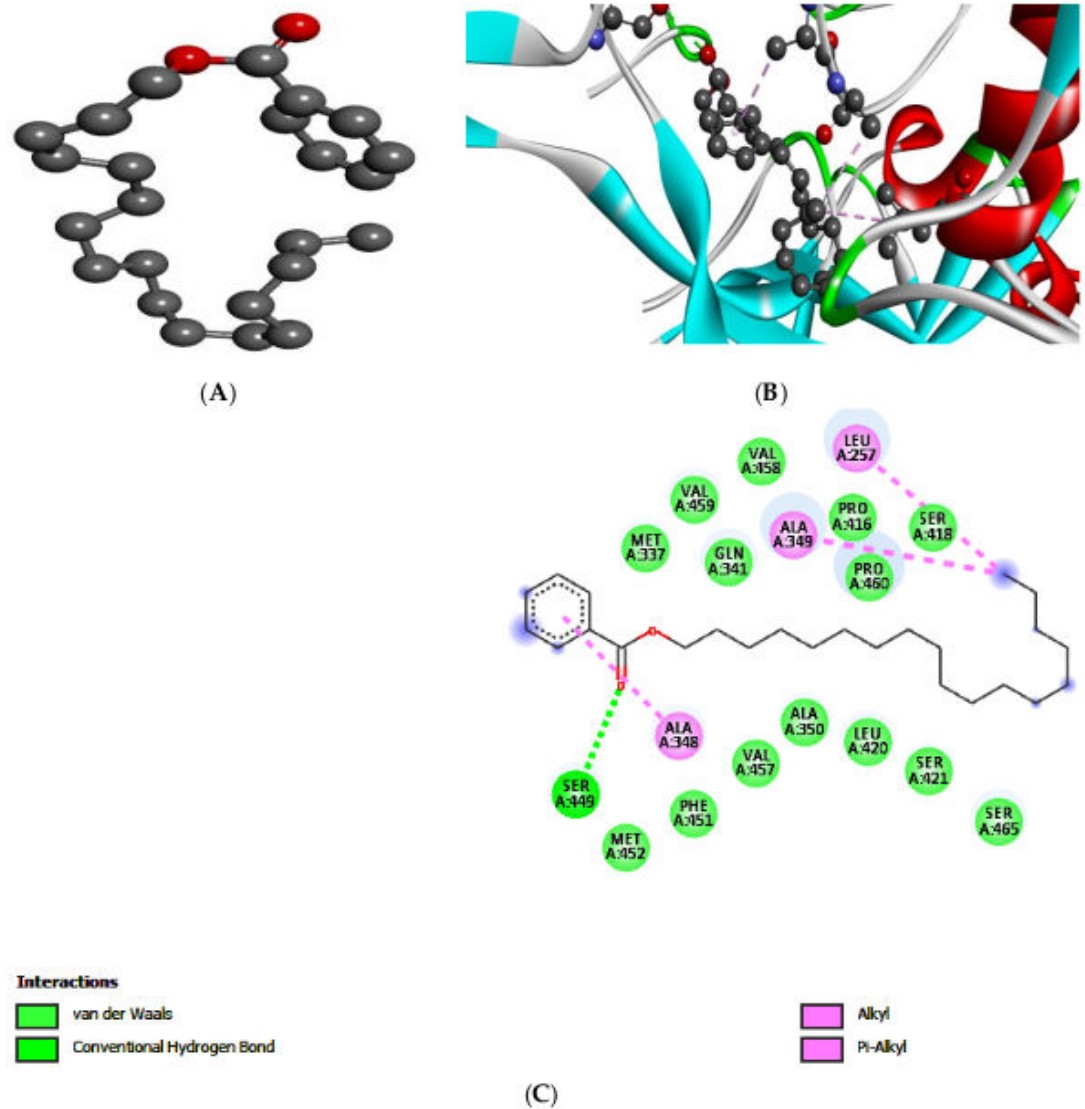

**Figure 4.** Interactions of legend (compound **2**) with *Leishmania tropica* promastigote surface gp63 enzyme, drawn through Discovery Studio Visualizer client 2016. (**A**) PDB structure and best pose of octadecyl benzoate; (**B**) General interaction of octadecyl benzoate with its targets; (**C**) Specific interaction of compound **2** with its targets.

Both compounds were evaluated against axenic cultures of *L. tropica* promastigotes. Test compounds exhibited dose-dependent (7.8–1000 µg/mL) anti-promastigote activity with $LD_{50}$ values of 10.40 ± 0.09 µg/mL (compound **1**) and 14.11 ± 0.11 µg/mL (compound **2**) after 72 h incubation using Glucantime and DMSO as positive and negative controls respectively (Figure 2). The $LD_{50}$ value of Glucantime was found to be 5.33 ± 0.07 µg/mL. *Leishmania* exists in two morphological forms, including promastigote and amastigote. Anti-leishmanial studies are usually performed at the promastigote stage because of the short culturing time [49]. The results of this study revealed promising anti-promastigote potentials for both compounds. In previous studies, methyl 3,4-dihydroxybenzoate has been isolated from *Asphodelus microcarpus*, demonstrating leishmanicidal activity ($LD_{50}$ = 33.2 µg/mL) against *L. donovani* promastigotes [15]. Furthermore, this compound has also been reported from *Piper glabratum* and *P. acutifolium* to exhibit strong leishmanicidal activity ($LD_{50}$ 13.8 µg/mL) against *L. amazonensis* [16]. To the best of our knowledge, this is the first report where both compounds are tested against *L. tropica*, which causes cutaneous leishmaniasis (CL), the prominent clinical form of leishmaniasis that is widespread in Pakistan [50]. Furthermore, we have extended our study in order to

discover the leishmanicidal preliminary mechanism, which will help in taking the current compounds to the next level of drug discovery.

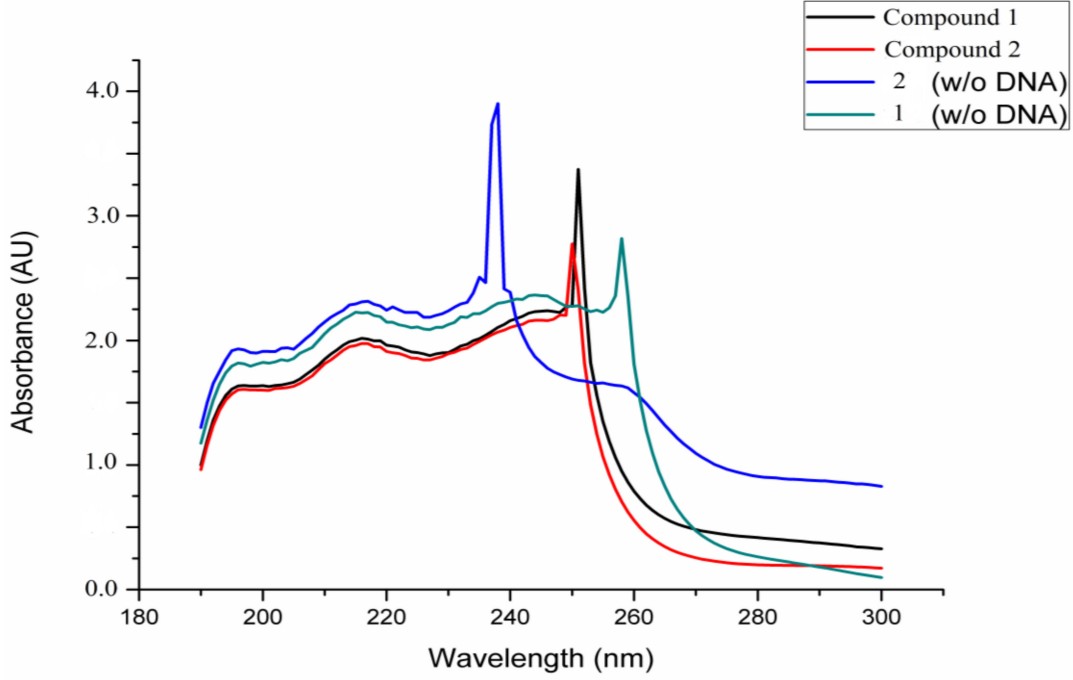

**Figure 5.** Compound **1** showed hyperchromism (blue shift), while compound **2** presented a hypochromism (red shift) after treatment with *Leishmania tropica* DNA.

The binding potential of compounds **1** and **2** against gp63 showed significant results. The compounds were assessed on the basis of binding affinity with gp63, in order to rationalize the anti-leishmanial potential in a qualitatively. The E-values for compound **1** and **2** were −5.3 and −5.6 kcal/mol, respectively. 2D interaction diagrams showing hydrogen bonds of compound **1** and **2** with gp63 are presented in Figure 3 and Table 1.

**Table 1.** E-value (kcal/mol) and post-docking analysis of best pose of compound **1** and **2** with GP63 enzyme.

| Targets | Compound 1 | | | Compound 2 | | |
|---|---|---|---|---|---|---|
| | **E-Value** | **H-Bonds** | **Bonding Residue** | **E-Value** | **H-Bonds** | **Bonding Residue** |
| **gp63** | −5.3 | 2 | Glu A:265 Leu A:344 | −5.6 | 1 | Ser A:449 |

The best-docked pose of ligand–protein complex, determined by minimum binding energy values, number of hydrogen bonds, and residues involved in hydrogen bonding, are summarized in Table 1. gp63 is a membrane-bound glycoprotein which binds to human natural killer cells—which are the first line of defense in numerous infections—and inhibits their proliferation, thus causing leishmaniasis [18]. Compound **1** was docked with the binding pocket of a gp63 receptor, and the results showed that a hydrogen bond was created between –OH of compound **1** with Glu265 residue. Likewise, another hydrogen bond was observed between $-CH_2$ and Leu344. Other multiple interactions were observed: pi–sigma interaction, pi–pi interaction, and pi–alkyl interactions of the benzene ring with Val261, His264, and Ala349, respectively. This showed that compound **1** bounded well to the gp63 receptor.

For compound **2**, one hydrogen bond was formed with the benzene ring and Ala348 residue, and pi–alkyl interactions were also observed for the carbonyl oxygen and methyl group ($-CH_2$) with Ser449 and Ala349 residues, respectively. The molecular-level interactions that showed binding with the gp63

involved major hydrophobic protein residues, thus, a drug with more lipophilic groups could more easily bind to gp63 and effectively inhibit its activity [38] (Figure 4). SYTOX green assay was carried out to check membrane damage, and visualization under the fluorescence microscope confirmed that both compounds were responsible for damaging the cell membrane of *Leishmania*. Full permeability (100%) was considered upon treatment with 0.1% Triton X-100, which was used as positive control. The *Leishmania* plasma membrane regulates the passage of nutrients, and the homeostasis, to maintain viability. SYTOX dye is a nucleic acid stain which combines with DNA, increasing the fluorescence up to 500-fold, which is observed as a green color [19,39,51].

The DNA interaction study was designed to check the interaction of compounds with *Leishmania* DNA. The spectrophotometric study showed significant interaction with DNA. Compound **1** showed an interaction with DNA through an increase in the absorption (hyperchromism) and a decrease in absorption wavelength (blue shift), while compound **2** exhibited a decrease in absorption (hypochromism) and an increase in absorption wavelength (red shift), thus confirming the interaction (Figure 5).

Apoptosis is a physiological procedure of cell suicide. It is an integral part of cell biology and occurs in both multicellular and unicellular organisms, including *Leishmania* promastigotes and amastigotes. Apoptosis greatly affects the survival of a parasite in a vector [21,22]. Acridine orange assays are used to characterize different stages of apoptosis in cells and to check for DNA damage. Acridine orange dye can insert into DNA and emits green fluorescence when bound to normal DNA or dsDNA, and red fluorescence when bound to damaged or ssDNA [38]. For both compounds, early apoptotic (EA) cells started appearing at 12 h and increased along with the treatment time, whereas late apoptotic (LA) cells were observed after 48 h of treatment. Compound **2** was not able to induce necrosis like compound **1**, which made compound **1** more effective towards killing *L. tropica* than compound **2**. EA cells had comparatively high motility to that of LA cells, whereas necrotic cells had either completely lost their motility or could only move very slowly. Our results showed that both compounds have apoptotic potential and caused death of *L. tropica* (Figure 6).

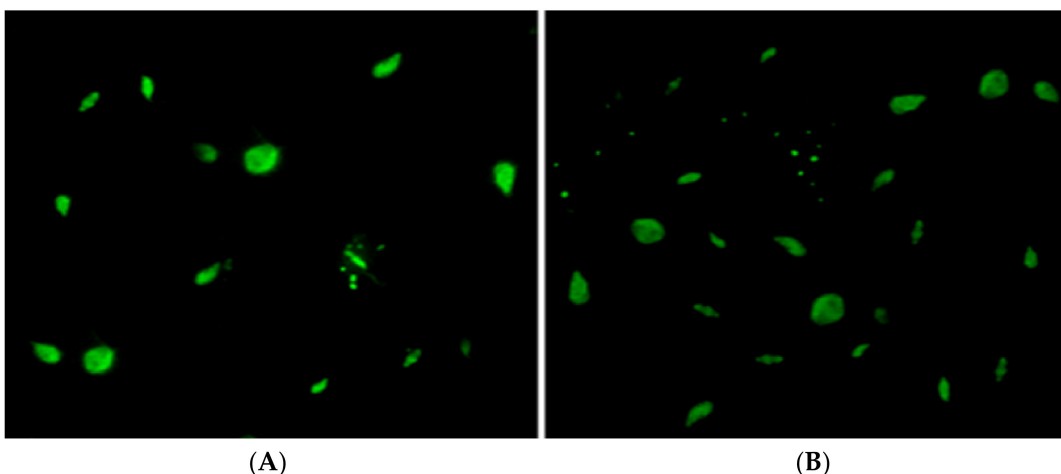

(**A**)　　　　　　　　　　　　　　　　　　　　　　　(**B**)

**Figure 6.** Apoptosis of *Leishmania tropica* cells treated with (**A**) compound **1** and (**B**) compound **2** using acridine orange fluorescent staining. Image after 48 h of treatment; dull green nuclei are indicative of apoptosis.

## 4. Conclusions

A literature survey showed that no plant-derived compound has been approved against leishmaniasis because the rate of isolation of pure compounds from extracts was low, and most of the research was confined to extracts/fractions of plants. Furthermore, some phytochemicals, like methyl curine, chondrocurine, and coumarin, have exhibited significant results against leishmaniasis, but no mechanistic or in vivo studies have been performed to date. This study aimed to isolate pure

compounds and carry out mechanistic studies to facilitate the development of cost-effective, safe, and efficacious drugs against leishmaniasis. The studied compounds exhibited cell death in *L. tropica* promastigote through apoptosis, which was further confirmed by DNA interaction and molecular docking studies. In our laboratory, further in vitro studies to evaluate both compounds against the amastigote form of *L. tropica* are ongoing. Future in vivo trials in rodent models are recommended, followed by determination of therapeutic dose.

**Supplementary Materials:** The following are available online at http://www.mdpi.com/2227-9717/7/4/208/s1.

**Author Contributions:** Conceptualization, S.M.S., F.U., A.N., M.A., A.S.; Methodology, S.M.S., A.N., N.U., S.H.; Software, A.-H.A.S., S.A.A.S., A.S.; Validation, F.U., I.U.; Formal Analysis, F.U., M.A.; Writing—Original Draft Preparation, S.M.S., M.A., N.U., I.U., F.U.; Writing—Review & Editing, M.A., F.U.; Supervision, F.U., A.N. All authors read and approved the final version of the manuscript.

**Funding:** This research received no external funding.

**Acknowledgments:** This work was partially funded by Higher Education Commission (HEC) of Pakistan and its support is gratefully acknowledged.

**Conflicts of Interest:** The authors declare no conflict of interest.

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
