# Peer review of "Benzoic Acid Derivatives of Ifloga spicata (Forssk.) Sch.Bip. as Potential Anti-Leishmanial against Leishmania tropica"

_processes, doi:10.3390/pr7040208_

Round 1
Reviewer 1 Report
The paper submissed to Processes journal and devote to the Special Issue "Extraction, Characterization and Pharmacology of Natural Products" focuses on the characterization of two potentials natural antileishmanials compounds based on benzoic acid, extracted from Ifloga spicata. Several experimental analysis and in silico calculations has been conducted to characterize the compounds and their actions against Leishmania Tropica.
In my opinion the paper deserves publication in Processes after the improvement of some concerns:
- how much is the purity of the samples?
- Did the authors consider the hypothesis to test the response of synthesized compounds 1 and 2 in order to compare it with the natural extracts ?
- Many drugs are used in the treatment of Leishmaniasis, but the only effective treatment is achieved with pentavalent antimonials. Can be interesting to compare the action of Sodium stibogluconate vs the two compounds studied here.
- Figures, especially NMR, are not decipherable and clear and must be drawn including some informations: (assignments of signals, labels on peak, their correlations..) necessary for the reader to understand the discussion during the MS.
- The resolution of the figures should be improved.
- English grammar should be improved by native English speaker
- I would add some more relevant references, especially more recent papers
Author Response
Authors response to Editorial and reviewers Comments
The authors cordially acknowledge the efforts and expertise of the associate editor and reviewers for considering our manuscript and providing us very useful suggestions. We have carefully addressed all the suggestions and we hope that the revised manuscript is significantly improved and hopefully will qualify the journal publishing standards.
Reviewer 1
Reviewer comments: The paper submissed to Processes journal and devote to the Special Issue "Extraction, Characterization and Pharmacology of Natural Products" focuses on the characterization of two potentials natural antileishmanials compounds based on benzoic acid, extracted from Ifloga spicata. Several experimental analysis and in silico calculations has been conducted to characterize the compounds and their actions against Leishmania Tropica. In my opinion the paper deserves publication in Processes after the improvement of some concerns:
Authors response:
Reviewer comments: how much is the purity of the samples?
Authors response: Both natural compounds were 100 percent pure as confirmed by performing NMR spectra by dissolving compounds in deuterated chloroform (CDCL3). Compounds 1 and 2 were soluble in it. Purity of the compounds was also checked by physical comparison of the isolated compounds 1 and 2 with commercially available pure samples and also checked their melting points. Compound 1 M. point was observed 134C0 which is exactly similar to pure commercially available Methyl 3,4-dihydroxybenzoate. Moreover comparative thin layer chromatography (TLC) was also carried out for both compounds with pure synthetic compounds and nearly same RF values were observed for both natural compounds.
Reviewer comments: Did the authors consider the hypothesis to test the response of synthesized compounds 1 and 2 in order to compare it with the natural extracts ?
Authors response: Many thanks for your nice suggestion. Yes we have already worked on this hypothesis. This is actually bioguided isolation of pure compounds from a traditionally used plant. Initially we performed studies on different bioactive natural extracts i.e n-hexane, chloroform and ethyl acetate after maceration of powered plant and screened all extracts for antileishmanial effect. We have submitted a manuscript on it to the Pakistan journal of botany and it is under review. Further we isolated these compounds from the most active fraction of the crude extracts and our results showed that pure compounds results were more significant as compared to natural crude extracts. However, we did not compared the activity of both compounds from natural and synthetic sources we only did it for isolated natural compounds.
Reviewer comments: Many drugs are used in the treatment of Leishmaniasis, but the only effective treatment is achieved with pentavalent antimonials. Can be interesting to compare the action of Sodium stibogluconate vs the two compounds studied here.
Authors response: Dear reviewer, many thanks for your constructive comment. Actually Leishmania is a tropical neglected disease and is a main issue of the under development countries. So drug development of this diseases is also negligible due to less target consumers and limited resources. Moreover, the main issue with Leishmaniasis is that all species respond differently to currently available drugs. We have used Glucantime as a standard because it is one the main drug used for Leishmania tropica in this region. Yes, we will use Sodium stibogluconate as suggested by the worthy reviewer in our near future studies but first of all we don’t have this drug here and also it is not used frequently in our region for the Leishmaniasis eradication.
Reviewer comments: Figures, especially NMR, are not decipherable and clear and must be drawn including some informations: (assignments of signals, labels on peak, their correlations.) necessary for the reader to understand the discussion during the MS.
Authors response: Dear reviewer, we have move the NMR data to supplementary file and provided its complete analysis as per kind suggestion. Many Thanks!
Reviewer comments: The resolution of the figures should be improved.
Authors response: All figures resolution improved to 350 dpi.
Reviewer comments: English grammar should be improved by native English speaker.
Authors response: Manuscript was language edited via a university Professor for technical and grammar mistakes. His suggestions were properly incorporated in the revised manuscript. Many Thanks!
Reviewer comments: I would add some more relevant references, especially more recent papers.
Authors response: More recent and relevant papers of years 2018 and 2019 were cited as suggested by worthy viewer.
Reviewer 2 Report
Ullah and Ayaz report a nice contribution on searching for anti-leishmanial potential drug targets. This manuscript presents an interesting piece in this field. All compounds here are meticulously characterized. This is an entirely suitable manuscript which could be published in processes after the following minor revisions.
References 4, 5, 6 and 36: absence of page number for the citing journals.
Author Response
Reviewer 2
The authors cordially acknowledge the efforts and expertise of the associate editor and reviewers for considering our manuscript and providing us very useful suggestions. We have carefully addressed all the suggestions and we hope that the revised manuscript is significantly improved and hopefully will qualify the journal publishing standards.
Comments and Suggestions for Authors
Reviewer comments: Ullah and Ayaz report a nice contribution on searching for anti-leishmanial potential drug targets. This manuscript presents an interesting piece in this field. All compounds here are meticulously characterized. This is an entirely suitable manuscript which could be published in processes after the following minor revisions.
Authors response: Dear reviewer may thanks for reviewing our manuscript and appreciating our efforts.
Reviewer comments: References 4, 5, 6 and 36: absence of page number for the citing journals.
Authors response: Thank you so much for highlighting this mistake. Page numbers were included in mentioned references.
Reviewer 3 Report
The anti-leishmanial potentials of benzoic acid derivates including Methyl 3-4 dihydroxy benzoate (Compound 1) and Octadecyl benzoate 27 (Compound 2) isolated from the plna Ifloga spicata (I. spicata) is reported in this manuscript. The compounds were characterized via FT-IR, mass spectrometry and multinuclear (1H and 13C) NMR spectroscopy. Anti-leishmanial potentials of the compounds were assessed against Leishmania tropica promastigotes KWH23.
The work is interesting. The ms is well written and it can be published in processes based on major revision
Line 99 “mg”
Abstract and throughout the text. The unit μg/μL is expressing ID50’s and not the IC50’s
Line 204 the uits are missing from ID50 value.
Figure 4 needs improvement. It seems that all measurements exhibit the same error.
Line 225 the “minimum” instead of the “maximum” energy shows the best binding position
figure 5 is in unacceptable quality as it is drawn. Its caption should be in the same form as the rest of the text.
Figure 7 what is the meaning of the negative absorbance? This figure needs to be prepared in better quality.
General comment. The figure captions should be uniform with the fonts used for the text.
Author Response
Reviewer 3
The authors cordially acknowledge the efforts and expertise of the associate editor and reviewers for considering our manuscript and providing us very useful suggestions. We have carefully addressed all the suggestions and we hope that the revised manuscript is significantly improved and hopefully will qualify the journal publishing standards.
Comments and Suggestions for Authors
Reviewer comments: The anti-leishmanial potentials of benzoic acid derivates including Methyl 3-4 dihydroxy benzoate (Compound 1) and Octadecyl benzoate 27 (Compound 2) isolated from the plna Ifloga spicata (I. spicata) is reported in this manuscript. The compounds were characterized via FT-IR, mass spectrometry and multinuclear (1H and 13C) NMR spectroscopy. Anti-leishmanial potentials of the compounds were assessed against Leishmania tropica promastigotes KWH23.
The work is interesting. The ms is well written and it can be published in processes based on major revision
Authors response: Dear reviewer may thanks for reviewing our manuscript and providing us highly useful suggestions. We have carefully considered all suggestions and revised the manuscript.
Reviewer comments: Line 99 “mg”
Authors response:
Reviewer comments: Abstract and throughout the text. The unit μg/μL is expressing ID50’s and not the IC50’s.
Authors response: Thanks for suggestion. IC50 is replaced by word LD50 throughout the abstract and text.
Reviewer comments: Line 204 the uits are missing from ID50 value.
Authors response: Thanks for highlighting this typo mistake. unit was put with LD50 value.
Reviewer comments: Figure 4 needs improvement. It seems that all measurements exhibit the same error.
Authors response: Figure 4 was improved to high resolution of 350dpi and also bar errors were corrected.
Reviewer comments: Line 225 the “minimum” instead of the “maximum” energy shows the best binding position.
Authors response: Thanks for highlighting this typo mistake. word maximum was replaced by word minimum.
Reviewer comments: figure 5 is in unacceptable quality as it is drawn. Its caption should be in the same form as the rest of the text.
Authors response: Figure 5 Quality was improved and also caption and text fonts were made uniform.
Reviewer comments: Figure 7 what is the meaning of the negative absorbance? This figure needs to be prepared in better quality.
Authors response: Dear reviewer, thanks for the constructive criticism. A negative absorbance mean if the sample is more transmitting than the reference. The spectral change process reflects the corresponding changes in DNA in its conformation and structures after the compound bound to DNA. Hypochromism (decrease in absorbance) results from the contraction of DNA in the helix axis and it was observed in case of compound 2 when interacted with leishmania DNA. Figure quality was improved to 350 dpi.
Reviewer comments: General comment. The figure captions should be uniform with the fonts used for the text.
Authors response: All figures captions and text of MS were made uniform.
Round 2
Reviewer 1 Report
The authors have partially solved my criticisms and I am not completely satisfied with their reply.
As far as from NMR spectra is clear that the sample is not 100% pure as they declare.
Several intense peaks are not assigned (Fig S1 ppm 7.25 and beetwen 0.75-2.5). The authors should explain.
Furthermore I don't see any improvements in the resolution of the figures.
Fig 3D , 4B and 5 I can't read the labels since them appear unfocused.
Moreover I suggest to mantain the right original proportion of the images.
I suggest to redraw the mosaic of images if is possible (Fig 3, 4).
For a print version I suggest to choice white background for the structures.
Please check again the number of the figures during the text, someone is referred to the previous version of the MS.
Author Response
Review 1 Report Form
Comments and Suggestions for Authors
Query: The authors have partially solved my criticisms and I am not completely satisfied with their reply. As far as from NMR spectra is clear that the sample is not 100% pure as they declare. Several intense peaks are not assigned (Fig S1 ppm 7.25 and between 0.75-2.5). The authors should explain.
Authors response: Dear reviewer, Thanks for pointing out the unlabelled peaks. As far as the unlabelled peaks are concerned, the peak at 7.28ppm is the solvent peak used as solvent, as mentioned in the spectra, labeled as CDCl3. This peak arises due to the residual solvent (CHCl3) in the deuterated (CDCl3) solvent. The other peaks (0.75-2.5ppm) are due to the solvents used for the extraction/isolation of the mentioned compound as explained in the manuscript (Methanol, Ethyl acetate, n-hexane). In order to confirm it please refer to the article publish in the Journal of organic Chemistry ( ACS Publication)(Hugo E. Gottlieb, Vadim Kotlyar, and Abraham Nudelman, NMR Chemical Shifts of Common Laboratory Solvents as Trace Impurities, J. Org. Chem., 1997, 62 (21), 7512-7515).
Query: Furthermore I don't see any improvements in the resolution of the figures.
Authors response: Thanks for pointing out. resolution improved for all figures.
Query: Fig 3D , 4B and 5 I can't read the labels since them appear unfocused.
Authors response: Figure 3D,4B and 5 redrawn. Many Thanks!
Query: Moreover I suggest to maintain the right original proportion of the images.
Authors response: Original proportion of images maintained. Many Thanks!
Query: I suggest to redraw the mosaic of images if is possible (Fig 3, 4).
Authors response: Dear reviewer, we did not got sufficient expertly in this and we improved the resolution of the images for better presentation. Many thanks!
Query: For a print version I suggest to choice white background for the structures.
Authors response: Background changed to white background. Many thanks!
Query: Please check again the number of the figures during the text, someone is referred to the previous version of the MS.
Authors response: Figures cross checked for correction. Many thanks!
Reviewer 3 Report
Although the majority of the comments done on the original version have been incorporated in its revised one, however there are some major to be done. Therefore the ms can be published in process after major revision.
Line 102 “600 mg”
Figure 3A and 3D are of very low quality. 3A: The molecular diagram has been resized asymmetrically between x and y axis. This results to not naturally drawn. 3D the letter cannot be written due to their small sizes.
Similarly Figure 4B
Figure 5 The x axis should intersect y axis at 0.0 and not at negative value.
Author Response
Reviewer 3
Comments and Suggestions for Authors
Query: Although the majority of the comments done on the original version have been incorporated in its revised one, however there are some major to be done. Therefore the ms can be published in process after major revision.
Query: Line 102 “600 mg”
Authors response: Dear reviewer, It is 600gram rather than 600 mg. Many Thanks!
Query: Figure 3A and 3D are of very low quality. 3A: The molecular diagram has been resized asymmetrically between x and y axis. This results to not naturally drawn. 3D the letter cannot be written due to their small sizes.
Authors response: Dear reviewer, as per your valuable suggestions Figures quality is improved. 3A redrawn and the best pose of docking is selected.3d redrawn. Many Thanks!
Query: Similarly Figure 4B
Authors response: Redrawn and Improved. Many Thanks!
Query: Figure 5 The x axis should intersect y axis at 0.0 and not at negative value.
Authors response: Thank you so much for pointing out. Figure 5 is redrawn and x axis intersect y at zero value.
Round 3
Reviewer 3 Report
The authors agree that 600 g extract (3.3%) were extracted from 18 kg of plant with methanol. However 600 gm is still given in the text. Therefore the ms can be published in process given that this correction will be done during proofs correction. accept with minor revision
.
Author Response
Reviewer response: The authors agree that 600 g extract (3.3%) were extracted from 18 kg of plant with methanol. However 600 gm is still given in the text. Therefore the ms can be published in process given that this correction will be done during proofs correction. accept with minor revision
Authors response: Dear reviewer many thanks for your kind suggestion. We have revised the manuscript as per your kind suggestion. Gm was replace by "g" throughout the text and highlighted green.